# Exploiting Environmental Variation to Improve Policy Robustness in Reinforcement Learning

## Abstract

Conventional reinforcement learning rarely considers how the physical variations in the environment (eg. mass, drag, etc.) affect the policy learned by the agent. In this paper, we explore how changes in the environment affect policy generalization. We observe experimentally that, for each task we considered, there exists an optimal environment setting that results in the most robust policy that generalizes well to future environments. We propose a novel method to exploit this observation to develop robust actor policies, by automatically developing a sampling curriculum over environment settings to use in training. Ours is a model-free approach and experiments demonstrate that the performance of our method is on par with the best policies found by an exhaustive grid search, while bearing a significantly lower computational cost.

## 1 Introduction

How does the agent's environment affect the robustness of policies it learns through deep reinforcement learning? Previous work has addressed the sensitivity of RL to changing visual environments by applying *domain randomization*, i.e. training on randomized visual environments (Tobin et al., 2017; Sadeghi et al., 2017), but there is limited work on the variability caused by the physics of the environment, such as object weight, surface friction, arm dynamics, etc. (Peng et al., 2018; Devin et al., 2017; Yu et al., 2017). In practical applications, one cannot assume the the task environment at test time has the same properties as the training environment, particularly if the training is conducted in simulation, and in most cases it is unfeasible to measure and categorize the changes. At the same time, enumerating all possible physical properties during training is extremely time-consuming and may still not lead to the best policy.

In this paper, we propose to develop RL algorithms that learn to exploit the training environment to discover the most robust and useful policy to handle any potential future environment, without needing explicit information about that future environment's underlying settings. We observe an interesting phenomenon, which we term '*inadvertent generalization*': training a task under certain physical settings results in more robust policies than others. For example, a pendulum policy trained on a low weight solves the task perfectly, but then fails on a higher weight setting. However, training on a single high weight leads to success on all lower weights. We observe forms of inadvertent generalization across multiple tasks - [i] simple pendulum inversion (with two different physics simulators), [ii] cart-pole balancing, [iii] ball-pushing (where the agent is required to push a ball to an arbitrarily specified goal). Curiously, intuition gained to explain the phenomenon on any one task does not necessarily extend to the others. We speculate that this may be because certain environments make the task harder and thus require more robust policies to solve the task.

Motivated by these observations, we propose to exploit variations in environments during training to learn a single policy that is capable of generalizing across environmental setting variations without incurring a significant additional cost, i.e. making the inadvertent generalization deliberate. We develop an approach called *Reward-guided Stochastic Curriculum* that automatically constructs a training curriculum for each learner to best motivate the learning of robust actor policies. We accomplish this by formulating the training curriculum as a multi-armed bandit problem, which seeks to maximize episodic rewards over all environmental settings of interest, utilizing separate

bandits for each environmental variable (where different physics properties correspond to different variables), thus ensuring linear growth with the introduction of additional variables, as opposed to polynomial. Experiments show that our method yields policies with performance similar to the best policies identified by an exhaustive grid search, while being computationally less expensive, since we only need to train one policy as opposed to one for each environmental setting.

## 2 PROBLEM DEFINITION

The primary focus of this paper is to characterize and exploit the phenomenon of inadvertent generalization so that we may better guide and utilize experiences gained to develop action policies that are more robust to environmental perturbation and variation. We build our analyses on the Deep Deterministic Policy Gradient (DDPG) algorithm (Lillicrap et al., 2016). DDPG is one of a class of actor-critic algorithms, where there exist separate actor and critic policies (each represented by a deep neural network). The critic attempts to learn the value of actions taken for a given state, which is then used to inform the improvement of the actor policy. We chose DDPG for its ability to handle continuous state and action spaces, which we consider to be important for both coarse and fine control, and for its status as a well-studied baseline algorithm with good stability and sample efficiency while learning (Duan et al., 2016).

We first motivate the problem by demonstrating inadvertent generalization on the inverted pendulum task. The aim of the pendulum task is to train an agent to hold a pendulum mass steady above its pivot point (as shown in Appendix A, Figure 2) . This is a task that can be considered solved by many of the existing deep-RL architectures (Schulman et al., 2017; Lillicrap et al., 2016; Wang et al., 2016; Wu et al., 2017; Schulman et al., 2015), but to the best of our knowledge, the robustness of RL policies to changes in the pendulum environment (or many classical control tasks) has not been well explored. We test whether an agent policy would still be capable of controlling the pendulum if the mass of the pendulum were changed. Our initial intuition was as follows: [i] decreasing the mass of the pendulum may result in significant overshoot of the vertical steady-point, potentially with a significant jitter in the motion, and [ii] increasing the mass might result in an inability to effectively swing-up the mass, resulting in failure. Both of these observations would make intuitive sense and could then be attributed to policies lacking an understanding of the environment dynamics. What we observed however is that, while hypothesis [ii] appears to hold, pendulum control policies trained with heavier masses generalized to a wider range of masses without a perceptible cost to stability. Table 1 provides a breakdown of the success rates of policies trained on individual OpenAI-gym Pendulum (Brockman et al., 2016) environment settings - in this case, with the only variable being the pendulum's mass. Observe that policies trained on heavier pendulums consistently outperform those trained on lower masses. It is important to note that all other aspects of the environment and action space were held equal - i.e. the maximum torque that the agent was allowed to apply was not changed and no other changes were made to how the physics in the environment are computed. To verify that these observations were not simply an anomaly of the physics simulator, tests were conducted on a separate pendulum environment built with the Unity game engine, which uses Nvidia's PhysX physics engine to simulate rigid-body dynamics. Our tests on the Unity pendulum environment show similar trends in behavior (as shown in Appendix C, Table 6), further establishing the

Table 1: *OpenAI-Gym Pendulum policy success rate (higher is better).* We evaluate multiple policies trained and tested on different OpenAI-Gym Pendulum-v0 environment settings. Rows represent performance for policies trained on a specific mass, columns correspond to specific test masses. Success rate is computed as the fraction of 8 trials with an average maximum deviation of less than 15 degrees, over 6 tests per test mass per trial, from the vertical steady point over the last 100 steps of a 300-step episode. Darker shading indicates worse performance.

| Train\Test | 1 | 2 | 3 | 4 | 5 | 6 | 7 | 8 | 9 | 10 | Avg |
|---|---|---|---|---|---|---|---|---|---|---|---|
| 2 | 0.75 | 0.63 | 0.50 | 0.25 | 0.38 | 0.38 | 0.38 | 0.25 | 0.13 | 0.13 | 0.38 |
| 4 | 0.75 | 0.88 | 0.75 | 0.75 | 0.75 | 0.63 | 0.50 | 0.38 | 0.38 | 0.13 | 0.59 |
| 6 | 0.88 | 1.00 | 1.00 | 1.00 | 1.00 | 0.88 | 0.88 | 0.75 | 0.38 | 0.25 | 0.80 |
| 8 | 0.75 | 1.00 | 1.00 | 1.00 | 1.00 | 1.00 | 1.00 | 1.00 | 1.00 | 1.00 | 0.98 |
| 10 | 1.00 | 1.00 | 1.00 | 1.00 | 1.00 | 1.00 | 1.00 | 1.00 | 1.00 | 1.00 | 1.00 |

Table 2: *Unity Ball-pushing policy error (lower is better).* Performance evaluation of multiple policies trained and tested on different custom Unity ball-pushing environment settings. Rows represent performance for policies trained on a specific ball mass, columns correspond to specific test masses. Errors are computed as the mean Euclidean distance of the ball from the goal evaluated on 6 separate trials, with 50 pre-defined tests per trial (to ensure fair comparison between policies).

| Train\Test | 2 | 4 | 6 | 8 | 10 | Avg |
|---|---|---|---|---|---|---|
| 2 | 0.21 | 0.21 | 0.22 | 0.22 | 0.22 | 0.22 |
| 4 | 0.40 | 0.23 | 0.25 | 0.25 | 0.25 | 0.28 |
| 6 | 0.66 | 0.37 | 0.36 | 0.36 | 0.37 | 0.43 |
| 8 | 1.25 | 0.66 | 0.63 | 0.63 | 0.64 | 0.76 |
| 10 | 1.15 | 0.54 | 0.46 | 0.45 | 0.46 | 0.61 |

effects of inadvertent generalization. Similarly, from training in the OpenAI-Gym CartPole-v1 environment (with minor modifications to support continuous control), we observe that training with heavier carts and poles seems to promote better generalization within the task, with cart masses having a stronger influence (see Appendix C, Table 7).

One might attribute these observations to the difference in physical inertia of pendulums, carts and poles of different masses - with a heavier pendulum presenting with higher inertia than a lighter one. This might further suggest that prioritizing high-inertia settings would result in improved generalization across environmental variations. Further testing with additional tasks however revealed that this is not the case. By analyzing the behavior of trained policies on a simple ball-pushing task (results presented in Table 2), where the agent is tasked with rolling a ball from an arbitrary starting position to an arbitrarily defined goal on a 2D plane, we observe that it is in fact the policies trained on lighter balls that do better at generalizing to variations in the environment when solving this task. Furthermore, policies trained with lighter balls also outperform their heavy-ball trained counterparts on test settings employing heavier balls. Independently, these observations can be rationalized to suggest that the higher inertia of the heavier balls requires the agents to exert more force on the ball to manipulate its position, resulting in a loss of finer control that might be developed with the lower-inertia (low mass) cases. However, it is clear that such intuition would in fact run counter to the behavior on the pendulum and cart-pole tasks.

Taken together, our observations and analyses demonstrate that there are classes of tasks, at least within the realm of continuous control, for which the following statement holds: given a task, there exists a window of generalizability for which training under a specific ideal set of environmental conditions results in a policy capable of generalizing to variations in the environmental settings. When the variations are limited, a grid search over the variants may be reasonable, but as the number of variables and the degree of variability increases, this quickly becomes impractical. This leads us to our **problem statement**: *we seek a principled approach that reliably and deliberately promotes the development of policies that are robust to environmental changes, similar to policies trained under 'ideal' settings, however, without the need for prior knowledge of the task/environment and without incurring significant additional computational cost over the cost of training a single policy.*

## 3 RELATED WORK

To the best of our knowledge, there have not been many explorations of robustness to the environment of a single task, as studied in this paper, however, we are able to draw insight from studies into multi-task reinforcement learning, specifically those where a single policy is trained to be utilized for all relevant tasks. Many of the existing single-policy methods appear to share a core idea: expose the agent to all the relevant task variations, attempt to account for different value associations that different tasks might encourage, and hope that some level of generalization might be achievable. Single-policy methods offer a key advantage towards generalization: they can be developed to function even when exact task specifications are not known. This was demonstrated in the performance of various approaches proposed/attempted in the OpenAI Gym Retro Challenge (Nichol et al., 2018), which was conducted in an attempt to systematically test algorithms abilities to generalize to unseen environments by having agents play levels of Sonic the Hedgehog. Interestingly, the best-performing approaches were achieved primarily by tuning the baselines "joint PPO", based

on the PPO (Schulman et al., 2017) architecture, and "joint Rainbow", based on Rainbow (Hessel et al., 2017). These "joint" networks were constructed with separate replay buffers for each trained level (task). During training, agents are exposed to a sampling of levels, and the gradients for each of these levels are averaged to produce an update to the universal policy, which is then used to update each level's agents. Similar approaches were taken by Model-Agnostic Meta-Learning (MAML) (Finn et al., 2017) and DeepMind's IMPALA (Espeholt et al., 2018). Key differences however are that they apply importance weighting to experience gained. MAML iteratively updates the policy in small steps relative to post-update losses of a series of proposed updates to the policy computed at the end of each episode (where each episode samples a new task). After sampling a set of tasks, MAML generates a final update to its primary policy from a scaled sum of the gradients generated from each proposed update. Instead of a direct averaging of agents, IMPALA applies their v-trace algorithm for importance weighting to compensate for the fact that their algorithm operates in a distributed manner with multiple (potentially un-synced) agents and learners, with some agents operating with outdated policies. IMPALA is primarily introduced as a distributed-learning framework, which conveniently possesses properties that allow it to be used in multi-task learning (where multiple agents can simultaneously train in different environments).

Yu et al. (2017) attempt to address a problem most similar to the one studied in this paper. The authors attempt to get around the problem of environmental variation by utilizing a separately trained online physics parameter(s) identifier to inform a joint 'universal' policy on the current state of the environment. However, there is a non-trivial cost associated with training the online identifier and the approach inherently limits itself to cases encountered in training (as demonstrated by their evaluations where the policy was not able to generalize beyond a certain threshold).

We also consider approaches that employ a sense of intrinsic motivation. While recent works in curiosity-driven learning achieve impressive results without any extrinsically defined reward signals from the environment (Burda et al., 2018), we focused on first addressing the more traditional RL problem structure with extrinsically defined rewards. One such approach to intrinsic prioritization of experience is implemented in Prioritized Experience Replay (PER) (Schaul et al., 2016), where experiences are sampled from the replay buffer proportionally to the magnitude of their Temporal-Difference (TD) error. A key inspiration for the method presented in this paper however comes from Graves et al. (2017), who tackle multi-task Natural Language Processing (NLP), utilizing a curriculum formulated as a multi-armed bandit. The advantage of this approach is that the curriculum can be automatically generated and would adapt to the experiences of individual learners, unlike the hand-tuned or goal-oriented curricula typically employed in RL. We borrow a similar idea, also building our curriculum on the Exp3 algorithm proposed by Auer et al. (2003), however, as discussed in Section 4, we employ a different valuation of the 'prediction gain'.

## 4 Reward-guided Stochastic Curriculum

As stated in Section 2, we seek to exploit the environment to find settings that lead to the most general policy for the task. We assume that all tasks have specific environment settings which can be controlled during training. The best policy is relatively straightforward to train when the ideal settings are known, however, these may not always be known a priori, and as the number of possible settings to consider increases, so too does the computational cost of exploring all variations of the settings in order to determine such settings.

Drawing inspiration from Graves et al. (2017), who tackle a multi-task NLP problem, we formulate the problem of developing an automated curriculum for learning generalization over environment settings for a given RL task as a multi-armed bandit problem, focused on minimizing regret and maximizing the actor's rewards. Each of the arms of the multi-armed bandit corresponds to an 'action' that the bandit can take, and each action would have a corresponding value (or payoff). The goal of the bandit is to maximize the payoff of every action, which would be trivial if the values of each arm is known, however, when action-values are not known, it is necessary to estimate the value by exploring the action space.

We define a curriculum as a sampling policy on the different environmental settings associated with a given task. A basic curriculum over $N$ possible environmental settings can be constructed as an $N$-armed bandit, with the syllabus of the developed curriculum intended to maximize the reward that the actor achieves over all the tasks. Over $T$ rounds of 'play', the bandit agent selects an action,

$a_t \in \{1, \ldots, N\}$, corresponding to a decision to train under a specific environment setting, and observes a payoff $r_t$, computed as the difference in mean rewards observed before and after training on the selected environment setting. The goal of the bandit/curriculum is to consistently select the settings which offer the best learning gains.

A key difference in our method from that employed by Graves et al. (2017) is in how we define the payoff, or the value gained by training on a specific setting. Graves et al. (2017) perform a comparison on the training loss before and after training, utilizing the same loss metric that is employed by the network. We instead compute our payoff based on the difference in mean episodic rewards before and after training. This choice was made based on an analysis of the Q values and TD errors of policies that generalized well on settings they were not trained on. We noted that, despite the 'good' performance of the actor, the critic was consistently wrong in its value predictions for any state/action pair, which was to be expected given that we do not model the physical environment. This led us to conclude that, in order to prioritize the performance of the actor policy, we would need to utilize a direct evaluation of the actor, which is reflected by the episodic rewards.

## 4.1 SINGLE VARIABLE ENVIRONMENTAL SETTING

To motivate the best choice of action that yields the lowest regret, we employ the Exponentially-weighted algorithm for Exploration and Exploitation (Exp3) (Auer et al., 2003). Specifically, we employ the Exp3.S variant of the algorithm to develop our multi-armed bandit's policy, which employs an $\epsilon$-greedy strategy and additively mixes weights to ensure that the probabilities of selecting any particular action is not driven to insignificance. We define $\epsilon$ to limit the maximum probability of any setting being selected. (*Note*: we present Exp3.S similarly to Graves et al. (2017), which is mathematically equivalent to the algorithm as it is presented in Auer et al. (2003)).

For a bandit policy defined by weights, $w_i$ for $i \in \{1, \ldots, N\}$, corresponding to the $N$ possible environment settings, at bandit-step $t$ and the bandit's action, $a_i$, the sampling probability, $\pi_t^{\text{Exp3.S}}(i)$ of action $i$ is given by:

$$a_i \sim \pi_t^{\text{Exp3.S}} \quad \rightarrow \quad \pi_t^{\text{Exp3.S}}(i) := (1 - \epsilon) \frac{\exp w_{i,t}}{\sum_j \exp w_{j,t}} + \frac{\epsilon}{N} \tag{1}$$

At the end of each bandit step, the weights are updated based on observed payoff, $r_t$:

$$w_{t+1,i} := \log \left[ (1 - \alpha_t) \exp \left( w_{t,i} + \hat{r}_{t-1,i}^{\beta} \right) + \frac{\alpha_t}{N-1} \sum_{j \neq i} \exp \left( w_{t,j} + \hat{r}_{t,j}^{\beta} \right) \right] \tag{2}$$

where $w_1 = 0$, $\alpha_t := t^{-1}$, and the importance sampled payoff is computed as:

$$\hat{r}_{t,i}^{\beta} := \frac{r_t \mathbb{I}_{[a_t=i]} + \beta}{\pi_t^{\text{Exp3.S}}(i)} \tag{3}$$

To bound the magnitude by which an arm's weight might change at any given step, payoffs, $r_t$, per bandit step, $t$ are scaled such that $r_t \in [-1, 1]$:

$$r_t := \begin{cases} -1 & \delta R_t < \mu_t^{20} \\ 1 & \delta R_t > \mu_t^{80} \\ \frac{2(\delta R_t - \mu_t^{20})}{\mu_t^{80} - \mu_t^{20}} - 1 & \text{otherwise} \end{cases} \tag{4}$$

where $\delta R_t = R_t - R_{t-1}$ is the true bandit policy payoff at step t, computed based on mean rewards achieved by the actor on the set of environment setting of interest, and $\mu^x$ represents the $x^{\text{th}}$ percentile of payoffs achieved: $\{r_s \ \forall \ s \leq t\}$.

## 4.2 MULTIPLE VARIABLE ENVIRONMENTAL SETTINGS

In addition to handling a single variable environment setting, we are also interested in efficiently handling environments where multiple settings might change - examples of single and multi-variable environments would respectively be the Pendulum environment, where the mass of the pendulum may

change, or the Cart-Pole environment, where the masses of both the cart and pole could change. Considering the combinatorial enumeration of all possible combinations of environment variables would cause the number of settings (and corresponding bandit arms) to grow by order $O(\prod_{m \in M} N_m)$. Instead, we propose a multi-multi-armed bandit solution, where a separate bandit is maintained for each variable. Crucially, this results in a linear growth in the number of arms to be maintained.

There are two primary differences between our approaches to the multi-variable and single variable settings for $M$ variables:

(i) The importance weighting of the scaled rewards (equation 3) is adjusted to account for the joint probabilities of the (assumed to be independent) variables:

$$\hat{r}^{\beta}_{M,t,i} := \frac{r_t \prod_{m \in M} \mathbb{I}_{[a_{m,t}=i_m]} + \beta}{\prod_{m \in M} {}^m\pi^{\text{Exp3.S}}_{m,t}(i_m)} \qquad (5)$$

where the pre-superscript $m$ reflects the actions and properties of the the $m^{\text{th}}$ bandit (corresponding to the $m^{\text{th}}$ environment variable.

(ii) The bandit weight policy weights are then updated per variable, effectively just as it was in equation 2, but using the importance sampled reward computed by equation 5:

$$w_{m,t+1,i} := \log \left[ (1-\alpha_t) \exp \left( w_{m,t,i} + \hat{r}^{\beta}_{M,t-1,i} \right) + \frac{\alpha_t}{N_m - 1} \sum_{j \neq i} \exp \left( w_{m,t,j} + \hat{r}^{\beta}_{M,t,j} \right) \right] \qquad (6)$$

---

**Algorithm 1:** Reward-guided curriculum for improving policy robustness

1 **Initialize**: $w_{m,i} = 0 \ \forall \, i \in N_m \ \forall \, m \in M$;
2 **for** $t = 1 \dots T$ **do**
3 $\quad$ Sample $M$ task-variable values $i_m$ under sampling properties defined by equation 1 for each of the $M$ systems of policy weights;
4 $\quad$ Sample $K$ task initializations uniformly from a valid space of initializations;
5 $\quad$ **for** $k \in K$ **do**
6 $\quad\quad$ Compute Initial Reward of actor-network policy $p_\theta$ on initialization $k$: $R^{pre}_k$;
7 $\quad$ **end**
8 $\quad$ Train network $p_\theta$ on $k \in K$;
9 $\quad$ **for** $k \in K$ **do**
10 $\quad\quad$ Compute Post-training Reward of network $p_\theta$ on initialization $k$: $R^{post}_k$;
11 $\quad$ **end**
12 $\quad$ Compute learning progress $\delta R_t := mean(\{R^{post}_k - R^{pre}_k\} \ \forall \ k \in K)$;
13 $\quad$ Map $\delta R_t$ to $[-1, 1]$ by equation 4;
14 $\quad$ Update weights $w_{m,i}$ by equation 6;
15 **end**

---

## 5 EVALUATION

Evaluations are conducted on the three task environments that were previously discussed: [i] Pendulum, [ii] Cart-Pole, and [iii] Ball-pushing. While the pendulum and ball-pushing environments have only a single variable environment setting (the pendulum and ball mass respectively), cart-pole has two variables (the pole mass and the cart mass), thus allowing us to evaluate the multi-variable version of our algorithm.

To meter the performance of our method, our results are compared against two key baselines: [i] The best results observed via a grid search (oracle) on policies trained exclusively on specific individual environment settings (i.e. the best inadvertently generalizing agent for each task, as presented in Tables 1, 2 and Appendix C, Table 7), and [ii] Policies trained under a joint/mixed training structure (joint), where the environment settings are varied every episode during training, with the episode settings drawn uniformly at random from a list of values of interest. This is similar to domain randomization. Additionally, for the pendulum task, which served as our primary sandbox for testing different ideas, we also provide comparisons against policies trained with Prioritized Experience

Replay (PER) and also policies where the curriculum's key payoff indicator was determined by the changes in TD-error instead of the episodic rewards but otherwise followed Algorithm 1. These baselines are motivated by initial attempts to guide training with respect to the TD-error, which seems reasonable given that it is the primary error metric for policy training in RL. Performance metrics for the Pendulum, Cart-Pole and Ball-pushing tasks are provided in Tables 3, 4 and 5 respectively.

Table 3: *Pendulum policy success rate comparisons (higher is better)*

| Test Mass | Policy | | | | |
|:---:|:---:|:---:|:---:|:---:|:---:|
| | Oracle | Joint | PER | TD-error-guided Curriculum | Reward-guided Curriculum (ours) |
| 1 | 1.00 | 1.00 | 1.00 | 1.00 | 1.00 |
| 2 | 1.00 | 1.00 | 0.93 | 1.00 | 1.00 |
| 3 | 1.00 | 1.00 | 1.00 | 1.00 | 1.00 |
| 4 | 1.00 | 1.00 | 1.00 | 1.00 | 1.00 |
| 5 | 1.00 | 1.00 | 0.93 | 1.00 | 1.00 |
| 6 | 1.00 | 1.00 | 0.79 | 0.85 | 1.00 |
| 7 | 1.00 | 1.00 | 0.64 | 0.77 | 1.00 |
| 8 | 1.00 | 0.93 | 0.57 | 0.69 | 1.00 |
| 9 | 1.00 | 0.86 | 0.29 | 0.69 | 1.00 |
| 10 | 1.00 | 0.57 | 0.14 | 0.46 | 0.93 |
| Avg | 1.00 | 0.94 | 0.73 | 0.85 | 0.99 |

Table 4: *Cart-Pole policy success rate comparisons (higher is better)*

| Cart Mass | Pole Mass | Policy | | |
|:---:|:---:|:---:|:---:|:---:|
| | | Oracle | Joint | Reward-guided Curriculum (ours) |
| 1.0 | 0.10 | 1.00 | 0.83 | 1.00 |
| | 0.25 | 0.83 | 0.83 | 1.00 |
| | 0.50 | 1.00 | 0.83 | 1.00 |
| | 1.00 | 0.83 | 0.83 | 0.93 |
| 3.0 | 0.10 | 0.83 | 1.00 | 0.86 |
| | 0.25 | 0.83 | 1.00 | 0.79 |
| | 0.50 | 0.83 | 0.83 | 0.79 |
| | 1.00 | 0.83 | 0.67 | 0.86 |
| 5.0 | 0.10 | 0.83 | 0.33 | 0.57 |
| | 0.25 | 0.83 | 0.33 | 0.64 |
| | 0.50 | 0.83 | 0.17 | 0.64 |
| | 1.00 | 0.67 | 0.17 | 0.64 |
| Avg | | 0.85 | 0.65 | 0.81 |

Table 5: *Ball pushing policy error rate comparisons (lower is better)*

| Test Mass | Policy | | | |
|:---:|:---:|:---:|:---:|:---:|
| | Oracle | Joint | PER | Reward-guided Curriculum (ours) |
| 2 | 0.21 | 0.55 | 4.85 | 0.44 |
| 4 | 0.21 | 0.64 | 4.77 | 0.44 |
| 6 | 0.22 | 0.68 | 4.71 | 0.45 |
| 8 | 0.22 | 0.68 | 4.66 | 0.45 |
| 10 | 0.22 | 0.69 | 4.65 | 0.45 |
| Avg | 0.22 | 0.65 | 4.72 | 0.45 |

It is immediately clear that our method outperforms policies built on joint sampling, PER (where tested) and the TD-error-guided curriculum (where tested), and achieves a performance closest to our oracle, of all the methods tested - with the Pendulum and Cart-Pole getting within 1% and 4% of their oracles' success rate respectively (noting that the Pendulum's oracle achieved a 100% success rate). In the case of the ball-pushing task, where we did not have a binary definition of success,

it can be noted that the average error is improved over joint sampling, being within $2\times$ of the oracle's error, as opposed to $3\times$. Our method also has a significantly lower computational cost than the oracle, needing to train only a single policy as opposed to $\prod_{m \in M} N_m$ policies. Additionally, as evidenced by Figure 1, the curricula developed appear to address the needs of each learner, adjusting the curriculum policies as necessary.

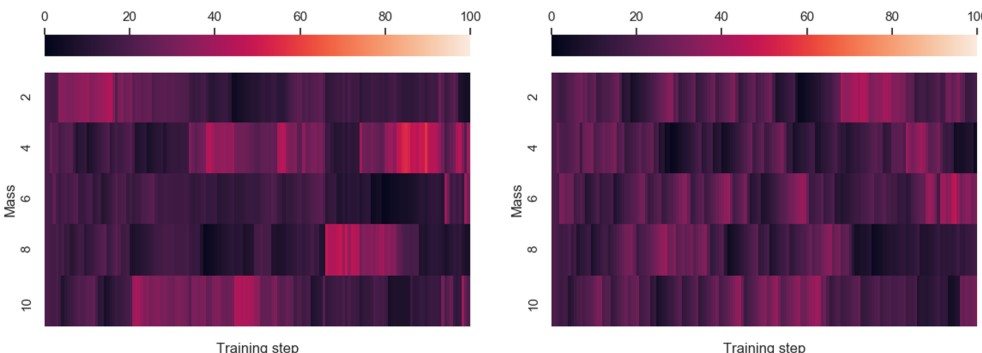

Figure 1: Cherry-picked comparison of two separate curricula evolution during training, represented as a heat-map to demonstrate probability distribution. Given the pseudo-random nature of episodic experience and policy-network training, it is important for curriculum to be able to adapt to the training experience. Note that Figure 1-left presents with a relatively clear sense of priority, initially favoring mass 2, then 10, then 4, then 6 and then 4 again. Contrast this with Figure 1-right, where there is no obvious pattern. Despite this, policies trained under both curricula present with equally successful performance, implying the ability of this training scheme to adapt consistently to different experiences when attempting to learn to solve a task. Note also that neither curriculum is uniform. Additional curriculum visualizations are provided in Appendix D.

It is interesting to note that TD error is apparently a bad metric for guiding curriculum choice and evolution. This is observed both with PER and the TD-error-guided curriculum - where the former samples the replay buffer proportionally to a transition's TD-error, and the latter adjusts the curriculum based on the TD-errors of transitions associated with previously tested settings. We hypothesize that prioritizing TD-error, which is inherently a measure of the next-state prediction capabilities of the critic, negatively impacts performance due to the fact that the critic is expected to always be wrong when working without a model of the environment and its settings - without explicitly knowing the current setting of the environment, it may simply be impossible to develop good predictions of expected future reward. Prioritizing TD-error as a metric by which to guide multi-setting (and possibly multi-task) learning may therefore wrongly bias policies towards minimizing the variance in TD-error across settings (or tasks). Rudimentary tests on the TD-errors on trained policies appear to support this hypothesis, however, due to time constraints, we have not been able to test this idea sufficiently to make a conclusive claim.

## 6    CONCLUSION

We proposed learning a stochastic curriculum, guided by episodic reward signals, to get the most out of an agent's environment and develop action policies robust to environmental perturbation. Furthermore, the curricula developed adapt to the experiences of each learner, allowing for a notion of self-reflection and self-correction. Not only does our method achieve performance close to the best policies found by an exhaustive grid search, it does so with a significantly lower computational cost, needing to train only a single policy, with minimal additional overhead, as opposed to $\prod_{m \in M} N_m$ policies. We also further demonstrate that neither uniformly sampling tasks, nor focusing on TD-error, as is common in multi-task RL, extends well to developing robust models for individual tasks. While our current approach is not designed to handle environments with sparse rewards or continuously varying settings, we hope to address these limitations with future work.

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

# A  TRAINING ENVIRONMENTS

We primarily use 4 training environments in all our experiments:

1. Pendulum-v0 from OpenAI Gym (Brockman et al., 2016), modified minimally to allow for programmatic control of the pendulum's mass. State space: $sin\theta$, $\cos\theta$, $\dot\theta$, Action space: $Torque$. Sample environment shown in Figure 2

2. Custom Unity Pendulum Environment - designed to provide a similar interface and response to the OpenAI Gym Pendulum implementation, however making use of Unity's built-in PhysX physics engine. State space: $\theta, \dot\theta$, Action space: $Torque$

3. CartPole-v1 from OpenAI Gym, modified to allow programmatic control of the pole and cart masses, as well as to be treated as a continuous control task, as opposed to one with discrete actions. State space: $x, \dot x$ (of cart), $\theta, \dot\theta$ of pole. Action Space: $Force$

4. Custom Unity Ball-pushing task. State space: $x_g, y_g$ position of goal, $x_b, y_b$ position of ball, $\dot x_b, \dot y_b$ velocity of ball

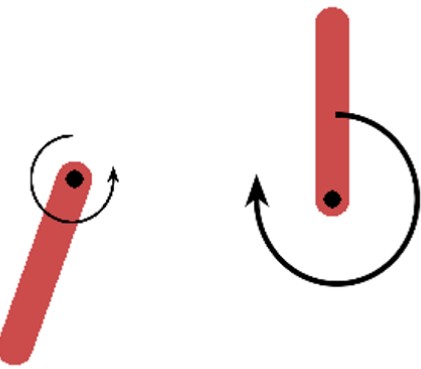

Figure 2: *Sample renders from Gym Pendulum-v0* (Left) Random Initialization, (Right) Successful Completion

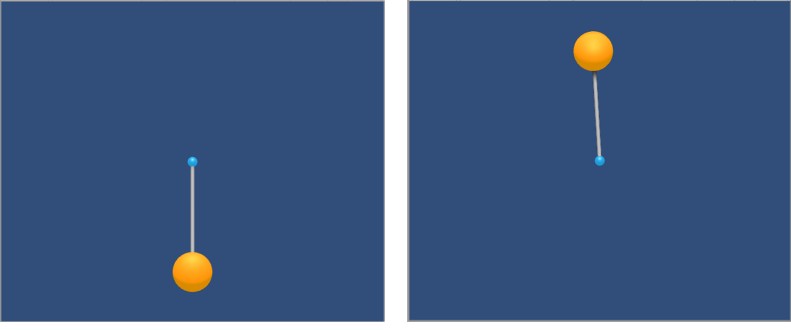

Figure 3: *Sample renders from Unity Pendulum* (Left) Random Initialization, (Right) Successful Completion

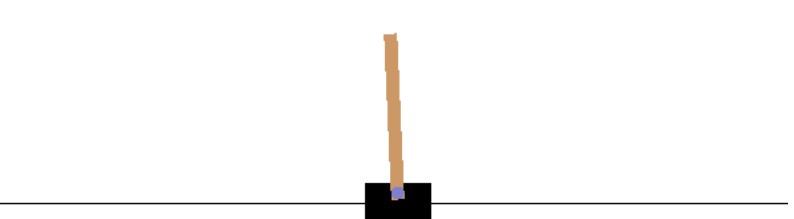

Figure 4: *Sample render from Gym CartPole-v1* Success is determined by maintaining the pole steady

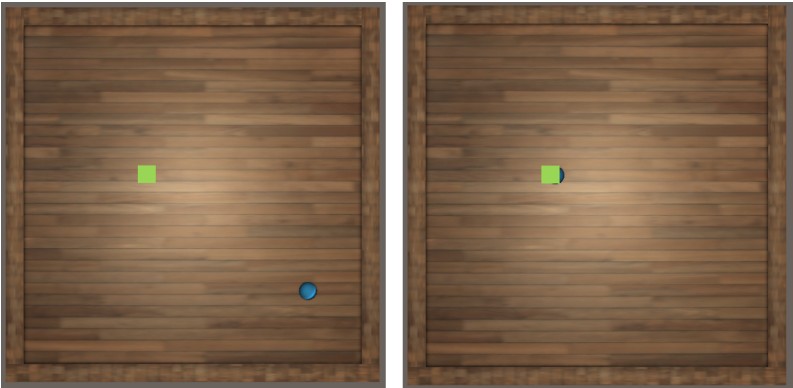

Figure 5: *Sample renders from Unity Ball-pushing* (Left) Random Initialization, (Right) Successful Completion

## B  EXPERIMENTAL DETAILS

Policy network configuration:

- **Network Architecture**: DDPG
- **Hidden layer configuration**: (400,300)
- **Additional notes**: Code adapted from Patrick Emami's code which is available on Github. Modifications were made to remove the use of tflearn and use only tensorflow. Additionally, the OpenAI's replay buffer code from their baselines  (Dhariwal et al., 2017) was adapted into this code to allow for the easy implementation of PER (Schaul et al., 2016)

Reward-Guided stochastic curriculum parameters:

- $\epsilon$: 0.05 for pendulum and ball-pushing, 0.2 for cart-pole
- $\beta$: 0.05 for pendulum and ball-pushing, 0.2 for cart-pole

## C  ADDITIONAL EXPERIMENTAL RESULTS

Table 6: *Unity-pendulum success rate for policies trained on individual environment settings (higher is better)*

| Train\Test | 1 | 2 | 3 | 4 | 5 | 6 | 7 | 8 | 9 | 10 | AVG |
|---|---|---|---|---|---|---|---|---|---|---|---|
| 2 | 0.83 | 0.83 | 0.50 | 0.17 | 0.17 | 0.00 | 0.00 | 0.00 | 0.00 | 0.00 | 0.25 |
| 4 | 1.00 | 1.00 | 1.00 | 1.00 | 1.00 | 1.00 | 1.00 | 0.67 | 0.83 | 0.67 | 0.92 |
| 6 | 1.00 | 1.00 | 1.00 | 1.00 | 1.00 | 1.00 | 1.00 | 1.00 | 0.67 | 0.67 | 0.93 |
| 8 | 1.00 | 1.00 | 1.00 | 1.00 | 1.00 | 1.00 | 1.00 | 1.00 | 1.00 | 1.00 | 1.00 |
| 10 | 1.00 | 1.00 | 1.00 | 1.00 | 1.00 | 1.00 | 1.00 | 1.00 | 1.00 | 1.00 | 1.00 |

Table 7: *Cart-pole success rate for policies trained on individual environment settings (higher is better)*. Note that the column and row headings contain the trained/tested pole and cart masses respectively within brackets

| Train/Test | (0.1,1) | (0.1,3) | (0.1,5) | (0.25,1) | (0.25,3) | (0.25,5) | (0.5,1) | (0.5,3) | (0.5,5) | (1,1) | (1,3) | (1,5) | AVG |
|---|---|---|---|---|---|---|---|---|---|---|---|---|---|
| (0.1,1) | 1.00 | 0.00 | 0.00 | 0.83 | 0.00 | 0.00 | 0.67 | 0.00 | 0.00 | 0.00 | 0.00 | 0.00 | 0.21 |
| (0.1,3) | 0.20 | 0.20 | 0.00 | 0.20 | 0.60 | 0.00 | 0.20 | 0.40 | 0.00 | 0.20 | 0.20 | 0.00 | 0.18 |
| (0.1,5) | 1.00 | 1.00 | 1.00 | 1.00 | 1.00 | 0.83 | 0.67 | 0.67 | 0.67 | 1.00 | 0.50 | 0.50 | 0.82 |
| (0.25,1) | 0.83 | 0.00 | 0.00 | 0.83 | 0.00 | 0.00 | 0.67 | 0.00 | 0.00 | 0.00 | 0.00 | 0.00 | 0.19 |
| (0.25,3) | 0.80 | 0.80 | 0.00 | 0.80 | 0.80 | 0.00 | 0.80 | 0.60 | 0.00 | 0.80 | 0.20 | 0.00 | 0.47 |
| (0.25,5) | 1.00 | 0.75 | 1.00 | 1.00 | 0.75 | 1.00 | 1.00 | 0.75 | 0.50 | 0.75 | 0.75 | 0.50 | 0.81 |
| (0.5,1) | 1.00 | 0.00 | 0.00 | 1.00 | 0.00 | 0.00 | 0.83 | 0.00 | 0.00 | 0.17 | 0.00 | 0.00 | 0.25 |
| (0.5,3) | 0.67 | 0.83 | 0.33 | 0.67 | 0.83 | 0.33 | 0.67 | 0.83 | 0.17 | 0.67 | 0.50 | 0.17 | 0.56 |
| (0.5,5) | 1.00 | 0.83 | 0.83 | 1.00 | 0.67 | 0.67 | 1.00 | 0.83 | 0.83 | 1.00 | 0.83 | 0.50 | 0.83 |
| (1,1) | 0.80 | 0.00 | 0.00 | 0.80 | 0.00 | 0.00 | 0.80 | 0.00 | 0.00 | 0.60 | 0.00 | 0.00 | 0.25 |
| (1,3) | 0.83 | 0.83 | 0.17 | 0.83 | 0.83 | 0.17 | 0.83 | 0.83 | 0.17 | 0.83 | 0.67 | 0.17 | 0.60 |
| (1,5) | 1.00 | 0.83 | 0.83 | 0.83 | 0.83 | 0.83 | 1.00 | 0.83 | 0.83 | 0.83 | 0.83 | 0.67 | 0.85 |

## D  PENDULUM CURRICULUM EVOLUTION

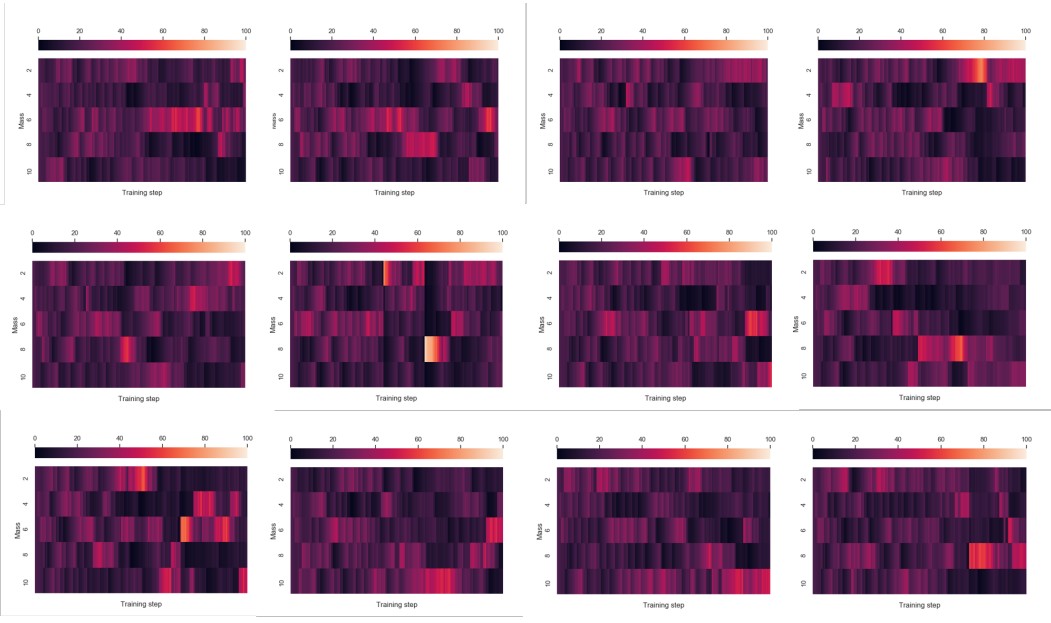

Figure 6: *Additional Sample Curricula for Pendulum training by Reward-guided Stochastic Curriculum*

