# OpenReview forum: "Exploiting Environmental Variation to Improve Policy Robustness in  Reinforcement Learning"
_ICLR.cc/2019/Conference_

### Official Review · AnonReviewer2 · 2018-11-05
**The papers proposes methods to robustify reinforcement learning algorithms against environment uncertainty which arises due to parametric variability. This is a interesting paper with promising results. What would make this paper a clear accept is the addition of experiments with high dimensional systems with more unknown parameters.**

**Rating:** 6
**Confidence:** 4

**Review:**

- Does the paper present substantively new ideas or explore an under-explored or highly novel question?

The paper claimed that there is limited work on the investigating the sensitivity of RL caused by the physics variations of the environment, such as object weight, surface friction, arm dynamics, etc. So the paper proposed learning a stochastic curriculum, guided by episodic reward signals (which is their contribution compared with previous related work) to develop policies robust to environmental perturbation.  Overall the combination of ideas is novel but the experimental results are limited in scope.

- Does the results substantively advance the state of the art?

The results advance the state of the art, since they are compared against : 1) the best results observed via a grid search (oracle) on policies trained exclusively on specific individual environment settings; 2) Policies trained under a mixed training structure, where the environment settings are varied every episode during training, with the episode settings drawn uniformly at random from a list of values of interest. Their 3 experiment results are competitive with 1) and much better than 2).

- Will a substantial fraction of the ICLR attendees be interested in reading this paper?

Yes, because the robustness of RL policies to changes in the physic parameters of the environment has not been well explored. Although previous investigations exist, and this paper’s algorithm is the combination of EXP3 and DDPG, it is still interesting to see them combined together to solve model uncertainty problem of RL with very good simulation results.

- Would I send this paper to one of my colleagues to read?

  I would definitely send the paper to my colleagues to read.


- In terms of quality:

Clear motivation; substantiated literature review; but the algorithms proposed are not novel and the question of whether the method will scale to more unknown parameters is not answered.

- I terms of clarity:

Easy to read.–Experimental evaluation is clearly presented.

- Originality:  The problem of developing an automated curriculum for learning generalization over environment settings for a given RL task is formulated as a multi-armed bandit problem, and EXP3 algorithm is used to minimize regret and maximize the actor’s rewards. Itis a very interesting application of EXP3, although such inspiration is drawn from a former multi-task NLP paper Graves et al. (2017).

- In terms of significance:

 The paper is definitely interesting and presents an  promising  direction. The significance is  limited because of the simplicity of the examples considered in the experimental session. It would be interesting to see how this method performs in problems with more states and more unknown parameters.

---

> ### Author Response · Authors · 2018-11-26
> **Response to AnonReviewer2**
>
> Thank you for your review.

---

### Official Review · AnonReviewer1 · 2018-11-06
**Curriculum design for dynamics randomization with a Bandits style method during training.**

**Rating:** 3
**Confidence:** 4

**Review:**

The paper looks at the problem of generalization across physical parameter varaition in learning for continuous control. The paper presents a method to develop a sampling based curriculum over env. settings for training robust agents.


* The paper makes an interesting observation on inadvertent generalization in robust policy learning.
However, the examples in both the cartpole and the pendulum cases seem not to be watertight.
For instance, the authors claim that
But from a dynamical system perspective in both cases, the controller is operating near limits.
The solution and subsequent generalization depend more on the topology of the solution space.
A heavy Pendulum is an overdamped system and required the policy to operate at the limits of action to generate momentum for swing up. Hence a solution for a lighter pendulum in implicitly included. Similarly, the rolling ball is an underdamped system, and where the policy operates near zero limits in light ball case to prevent the system from going unstable. Adding mass results in damping which makes it easier. In this case, as well the solution space is implicitly contained.


But this is not a novel observation. Similar observations have been made for Robust control and Model-Reference Adaptive Control.
The paper also overlooks a number of related works in model-free randomization [4], adaptive randomization [3], adversarial randomization [5,6]. The method also does not compare with model-based methods for adaptive policy learning and iLQR based methods to handle this problem [2, 7].


The argument that the method is model-free is perhaps not as acceptable since the model parameters need to be known apriori for adaptation. The policy itself may be model-free but that is a design choice.
A good experimental evaluation for this is generalization across known unknowns and unknown unknowns.


* The algorithm itself is reasonable but the problem setup and choice of a discrete dynamics parameter choices are questionable. The bandit style method operates over a discrete decision set.
It also assumes in the multi-parameter setting that they are independent, which may not be true very often.

The algorithm proposed itself isnt novel, but would have been justified if the results supported the use of such a method.

* Experiments are quite weak.
Both the experimental domains are rather simplistic with smooth nonlinear dynamics. There are more sophisticated and interesting continuous control environments such as control suite [1] or manipulation suite [2].

It would be useful to see how tis method works in more complicated domains and how the performance compares with simpler methods such as joint brute-force randomization both in performance and in computation.

Questions:
1. Please provide details of Algorithm 1. How are the quantities K and M related?
2. What is the process of task initialization? What information is required and what priors are used. Uniform prior over what range?


In summary, the authors explore an interesting adaptive curriculum design method. However, in its current form, the work needs more thought and empirical evaluation for the sake of completeness.


References:
1. Model Reference Adaptive Control [https://doi.org/10.1007/978-1-4471-5102-9_116-1
]
2. ADAPT: Zero-Shot Adaptive Policy Transfer for Stochastic Dynamical Systems [https://arxiv.org/abs/1707.04674]
3. EPOpt: Learning Robust Neural Network Policies Using Model Ensembles [https://arxiv.org/abs/1610.01283]
4. Domain Randomization for Transferring Deep Neural Networks from Simulation to the Real World
[https://arxiv.org/abs/1610.01283]
5. Certifying Some Distributional Robustness with Principled Adversarial Training [https://arxiv.org/pdf/1710.10571.pdf]
6. Adversarially Robust Policy Learning: Active Construction of Physically-Plausible Perturbations [http://vision.stanford.edu/pdf/mandlekar2017iros.pdf]
7. Synthesis and Stabilization of Complex Behaviors through Online Trajectory Optimization [https://homes.cs.washington.edu/~todorov/papers/TassaIROS12.pdf]

---

> ### Author Response · Authors · 2018-11-26
> **Response to AnonReviewer1**
>
> Thank you for your review.
>
> --------------------------------
> I) Response to your questions:
>
> 1. K and M are unrelated. 'M' is the total number of environment settings can be changed. K is the number of 'tasks' initialized under a specific environment configuration - i.e. after sampling some settings, we train on those settings for K episodes without changing them
>
> 2. Tasks are initialized uniformly over the state space of the task. Environment settings are initialized per the sampling policy defined by the bandit.
>
> --------------------------------
> II) Please clarify:
>
> In your review, you have a sentence, which seems to end abruptly: "For instance, the authors claim that". Could you please clarify what you meant to say?
>
> You mention “more sophisticated and interesting continuous control environments such as control suite [1] or manipulation suite [2]”, however neither references [1] nor [2] in your review seem to address such suites. Could you please clarify which suites you are referring to?
>
> --------------------------------
> III) Primary contributions of this work and novelty:
>
> Primarily, we sought to introduce a learning scheme that would combat the problem of policy brittleness in RL when policies are exposed to environmental behavior not seen in training. We recognize that work has been done in model-based RL to combat these problems, however, we specifically focus on the model-free setting.
>
> We focus specifically on a model-free formulation because measuring the properties of a test environment may sometimes be impossible or infeasible - for example, in a real-world task, adding sensors and processing to perceive the properties of the environment may prove too expensive to be feasible. If the action policy could instead be robust to changes, the problem is somewhat alleviated.
>
> To your point about model parameters needing to be known a priori during training, we would like to note that we also test our trained policies against environmental settings that were not explicitly trained for. For example, as shown in Table 3, we train the pendulum with only masses 2, 4, 6, 8 and 10, but masses 1, 3, 5, 7 and 9 are also tested and appear to be implicitly handled. We observed similar trends with ball-pushing. The apparent efficacy demonstrated by this result also seemed to justify the choice of discrete bandit scheme.
>
> When discussing inadvertent generalization, we were specifically referring to the observation that the brittleness of RL policies seems to change quite significantly, and is even seemingly mitigated, in response to small changes in the environment during training, in ways that allow it to remain robust despite lacking a model of the environment in test time. To the best of our knowledge, this has not been addressed in prior work.
>
> --------------------------------
> IV) Addressing related works:
>
> 1. We note that model-free randomization (reference [4] in your review) is effectively uniform random sampling employed during training - which is equivalent to the ‘joint’ training, which we use a baseline.
>
> 2. Adaptive randomization (reference [3]) appears to adapt the training distribution by testing policies in a target environment . However, in our work, we do not assume access to tests on target domains during training, disallowing us from adopting a similar adaptation approach.
>
> 3. We sought to demonstrate our algorithm on a few controls tasks, as is the current norm in deep RL, where it was tentatively more intuitive to understand the control schemes and strategies. However, we view our approach as more generally applicable to deep RL and thus did not focus on comparing it against controls techniques like MRAC, MPC or LQR. Unlike traditional controls, our method is directly extensible to tasks where variations in the task environment are not limited to changes in control system dynamics - examples include training object-detection/recognition to be more robust to scene lighting, or game AI to be more robust to level design.

---

### Official Review · AnonReviewer3 · 2018-11-07
**An interesting view point on robustness.**

**Rating:** 5
**Confidence:** 3

**Review:**

This paper investigated the robustness of RL policies learning under different environmental conditions.

Based on the observations that policies learnt in different experimental settings lead to different generalizability, the authors proposed an EXP3 based reward-guided curriculum for improving policy robustness. The algorithm was tested on inverse pendulum, cart-pole balancing, and ball-pushing in OpenAI gym.

The paper is well-organized and easy to understand. Written errors didn't influence understanding. Papers in the references were not properly cited.

It is an interesting discovery that different environment brewed different policies with different robustness/generalizability in daily life. However, these are also easily derivable in physics, especially in the three experiments tested in the paper. It would be more complete to compare with PID controllers.

---

> ### Author Response · Authors · 2018-11-26
> **Response to AnonReviewer3**
>
> Thank you for your review.
>
> Citation issues have been resolved in the latest revision of the paper.
>
> While a PID controller is likely to solve the problems we addressed, our focus was not on being able to implement a control policy for individual tasks, but rather to demonstrate a technique to combat policy brittleness in generic deep RL - where learned policies fail when presented with data from domains different to those they were trained on, which is often a problem in machine learning.

---

### Meta-Review · Area_Chair1 · 2018-12-14
**The paper needs improvement**

**Confidence:** 4
**Recommendation:** Reject

**Metareview:**

The paper presents a strategy for randomizing the underlying physical hyper-parameters of RL environments to improve policy's robustness. The paper has a simple and effective idea, however, the machine learning content is minimal. I agree with the reviewers that in order for the paper to pass the bar at ICLR, either the proposed ideas need to be extended theoretically or it should be backed with much more convincing results. Please take the reviewers' feedback into account and improve the paper.